# Peaceful dying among Canada's elderly: An analysis of the Canadian Longitudinal Study on Aging

Komal Aryal[1]*, Aaron Jones[1,2], Peter Tanuseputro[3], Lauren E. Griffith[1], Paul C. Hebert[4], Susan Kirkland[5], Deborah J. Cook[1], Andrew P. Costa[1,2]

1 Department of Health Research Methods, Evidence, and Impact, McMaster University, Hamilton, Ontario, Canada, 2 St. Joseph's Health System Centre for Integrated Care, Hamilton, Ontario, Canda, 3 Department of Family Medicine and Primary Care, University of Hong Kong, Hong Kong, 4 Department of Epidemiology and Public Health, University of Ottawa, Ottawa, Canada, 5 Department of Community Health and Epidemiology, Dalhousie University, Halifax, Canda

* aryalk@mcmaster.ca

**Data Availability Statement:** Data cannot be shared publicly because of privacy concerns (i.e., sensitive patient information). Data are available from the Canadian Longitudinal Study on Aging for

## Abstract

### Introduction

Death is universal, yet relatively little is known about how Canadians experience their death. Using novel decedent interview data from the Canadian Longitudinal Study on Aging we describe the prevalence and characteristics of peace with dying among older Canadians.

### Methods

We conducted a secondary analysis of decedent interview data from the Canadian Longitudinal Study on Aging. Proxies of deceased Canadian Longitudinal Study on Aging participants reported on participants' end-of-life experiences between January 2012 to March 2022. We examined end-of-life characteristics and their association with proxy reports of experiencing peace with dying. We conducted regression analysis to explore the association between demographic and end-of-life characteristics and experiencing peace with dying.

### Results

Of 3,672 deceased participants, 1,287 (35.0%) had a completed decedent questionnaire and were included in the analysis. Respondents reported that two-thirds (66.0%) of the deceased experienced peace with dying and 17% did not experience peace with dying. The unadjusted odds of experiencing peace with dying were higher for those with an appointed power of attorney (OR 1.80; CI 1.39–2.33), those who died of cancer (OR 1.71; CI 1.27–2.30), those in hospice/receiving palliative care (OR 1.67; CI 1.19–2.37), individuals older than 75 years (OR 1.55; CI 1.04–2.30), or widowed (OR 1.53; CI 1.12–2.10). Widowhood (OR 1.51; CI 1.01–2.29), having an end-of-life SDM (OR 1.58; CI 1.14–2.17), and dying of cancer (OR 1.67; CI 1.19–2.23) increased the adjusted odds of dying with peace.

researchers who meet the criteria for access to confidential data (https://www.clsa-elcv.ca/data-access/).

**Funding:** : Funding for this study was obtained from the Canada Research Chair in Integrated Care for Seniors. The funders had no role in study design, data collection and analysis, decision to publish, or preparation of the manuscript. This research was made possible using the data/biospecimens collected by the Canadian Longitudinal Study on Aging (CLSA). Funding for the CLSA is provided by the Government of Canada through the Canadian Institutes of Health Research (CIHR) under grant reference: LSA 94473 and the Canada Foundation for Innovation, as well as the following provinces: Newfoundland, Nova Scotia, Quebec, Ontario, Manitoba, Alberta, and British Columbia. This research has been conducted using the CLSA (Baseline Tracking Dataset version 4.0, Baseline Comprehensive Dataset version 7.0, Follow-up 1 Tracking Dataset version 3.0, Follow-up 1 Comprehensive Dataset version 4.0, Follow-up 2 Tracking Dataset version 1.0, Follow-up 2 Comprehensive Dataset version 1.0, and the Decedent Questionnaire Dataset version 1.0) under Application 2209016. The CLSA is led by Drs. Parminder Raina, Christina Wolfson and Susan Kirkland. Andrew Costa is supported by the Canada Research Chair in Integrated Care for Seniors. Aaron Jones is supported by the Schlegel Chair in Clinical Epidemiology & Aging. Lauren Griffith is supported by the McLaughlin Foundation Professorship in Population and Public Health. Deborah Cook is supported by the Canada Research Chair in Knowledge Translation in Critical Care.

**Competing interests:** The authors have declared that no competing interests exist.

## Conclusions

Close to 1 in 5 older Canadians may not experience peace with dying, which supports greater focus on improving the end-of-life care. Our findings suggest that advanced planning may enhance the experience of a peaceful death in Canada.

## Introduction

Death holds profound significance as an inevitable event that marks the culmination of life's journey. Achieving a good quality of death is heavily influenced by the ability to experience peace with dying [1, 2]. The concept of dying peacefully can involve physical comfort, emotional acceptance, and spiritual well-being [3, 4]. From 2000 to 2013, the overall quality of death and dying experiences declined, highlighting the urgent need to understand and potentially improve the quality of the dying process [5].

With Canada's population aged 85 and older projected to double by 2050, alongside a 25% increase in overall deaths, understanding the factors that influence peace with dying is crucial to improve the overall dying experience [6, 7]. Religious beliefs, healthcare professionals' attitudes, and end-of-life care practices directly shape a person's experience of peace with dying [8–10]. Significant pain during end-of-life or receiving minimal support that does not align with personal preferences, has been linked to not experiencing peace with dying [11, 12]. Despite the importance of dying with peace and dignity for individuals, caregivers, and healthcare providers [13], few population-based data sources in Canada explore the relationship between end-of-life characteristics and peace with dying [14–16]. Identifying end-of-life characteristics associated with a peaceful death may allow older adults to experience a good quality of death and enable targeted improvements in care when these factors are not met.

Using newly released data from the Canadian Longitudinal Study on Aging we sought to describe the end-of-life characteristics associated with proxy reports of decedents' experience of peace with dying. Based on previous findings, we hypothesized that personal and end-of-life characteristics, such as dying of cancer is correlated with experiencing peace with dying, while unexpected or unplanned deaths, such as those due to chronic illnesses or lack of an appointed decision-maker, decrease the likelihood of experiencing peace.

## Methods

### Study design and data source

We conducted a secondary analysis of decedent interview data from the Canadian Longitudinal Study on Aging (CLSA). The CLSA is a prospective cohort study platform with a national, stratified sample of 51,338 community-dwelling middle-aged and older adults, aged 45–85 years at baseline who are followed every 3 years. Previous reports have described the CLSA's design and methodology [17–19]. To summarize, the CLSA comprises of the Tracking cohort, which includes participants randomly selected from all 10 Canadian provinces, and the Comprehensive cohort, which includes participants randomly selected from within a 25–50 km radius of one of 11 data collection sites located in British Columbia, Alberta, Manitoba, Ontario, Quebec, Nova Scotia, and Newfoundland. Both cohorts collect similar data, while the Comprehensive cohort also undergoes a more detailed physical assessments. Participant demographic and social characteristics at baseline are comparable to the 2011 Canadian census [17].

### Decedent questionnaire

Next of kin or primary contact (most often identified by the CLSA participant at baseline) were contacted by mail once the CLSA received the confirmation of death from a family member, friend, or through provincial death records. Next of kin or primary contacts were then contacted via telephone two weeks after mailing to invite participation in a decedent interview. Decedent questionnaires (available online) [20] were completed between January 2012 to March 2022 by telephone interview (French or English) with a trained CLSA decedent interviewer. Interviews were completed 2 years after death on average (ranging between 10 days to 6.3 years), depending on respondent availability and release of the participant's deceased status.

### Participants

CLSA participants who died between from January 6, 2012, to March 15, 2022, and for whom we had a completed decedent interview were included for analyses.

### Peace with dying

Respondents were asked whether they believed the deceased participant experienced peace with dying, defined as being at peace with the concept of dying during the last week of their life. There were six possible response options: 1) yes, they experienced peace with dying, 2) they were 'somewhat' at peace with dying, 3) they did not experience peace with dying, 4) this question is not applicable, 5) they don't know if the deceased participants experienced peace with dying, and 6) refused to answer the question. We used 1) yes, they experienced peace with dying, as the dependent variable in the analyses.

### Statistical analysis

We examined participant and end-of-life characteristics from the decedent questionnaire, including the location of death, cause of death, arrangements for health care decision-making, and arrangements for end-of-life care decision-making. Descriptive summaries were generated to characterize participant and end-of-life characteristics and compare between participants with a complete and incomplete decedent interview. We also compared the participant sociodemographic characteristics from both the CLSA and Statistics Canada for deceased Canadians.

   We used a correlation matrix and computed the variance inflation factor (VIF) to examine multicollinearity among end-of-life and participant characteristics. We performed unadjusted logistic regression to estimate associations with peace with dying and multivariable logistic regression to determine adjusted associations and adjust for any potential confounders. We reported the area under the receiver operating curve to assess model discrimination. We performed a subgroup analysis by sex, between those dying of cancer compared to those dying of other causes, and between time of death to time of decedent interview, to examine differences in the associations with dying with peace.

### Ethics approval

This secondary analysis was approved by the Hamilton Integrated Research Ethics Board (2023-16023-C).

## Results

There were 3,672 CLSA participants who died between 2012–2022, and 1,287 (35.0%) had a completed decedent questionnaire. CLSA decedents with a completed decedent questionnaire

were on average 73.6 year of age at death, 62.7% were married or in a common-law relationship, 39.7% died of cancer, and 49.0% died in hospital. Deceased CLSA participants with a completed decedent interview were more likely to be male, older, married, and identify a religious affiliation compared to those without a decedent interview (**Table 1**). CLSA deceased participants were older at death, more likely to be married or in a common-law relationship, less likely to have died of cancer, and less likely to have died in hospital compared to the general Canadian population who were 45 years or older [21] and died between 2012–2022 (**S1 Table**).

Approximately two-thirds (66.0%, n = 855) of deceased participants experienced peace with dying, 7.0% (n = 85) somewhat experienced peace with dying, and 1 in 5 participants (17.0%, n = 213) did not experience peace with dying (**Fig 1**). Almost two-thirds (62.0%, n = 798) of deceased participants were male, 66.7% (n = 858) were 75 years old or older, 62.7% (n = 807) were married, and 35.1% (n = 452) had a bachelor's degree or higher. Regarding end-of-life characteristics, 39.7% (n = 511) died of cancer, 49% (n = 631) died in hospital, and 75.1% (n = 967) and 81.5% (n = 1,049) of participants had appointed proxies responsible for deciding any end-of-life choices known as end-of-life substitute decision makers (SDM) and health care SDMs, proxies responsible for broader healthcare decisions, respectively (**Table 2**). Based on the results of the correlation matrix and variance inflation factor, these characteristics were not significantly correlated (**S2 and S3 Tables**).

Our unadjusted analysis shows that being 75 years old or older (OR 1.55; CI 1.04–2.30), widowed (OR 1.53; CI 1.12–2.10), having Activities of Daily Living (ADL) or Instruments of Daily Living (IADL) impairment (e.g., moderate, OR 1.66; CI 1.13–2.48), having an appointed end-of-life SDM (OR 1.80; CI 1.39–2.33) or healthcare SDM(OR 1.63; CI 1.22–2.18), dying of cancer compared to heart disease (OR 1.71; CI 1.27–2.30) and dying in hospice (OR 1.67; CI 1.19–2.37), were associated with experiencing peace with dying (**Table 2**). Adjusted odds showed that being widowed (OR 1.51; CI 1.01–2.29), having an end-of-life SDM (OR 1.58; CI 1.14–2.17), and dying of cancer (OR 1.67; CI 1.19–2.23) increased the odds of dying with peace. Model discrimination for the adjusted model was fair (AUC = 0.65) and similar to the unadjusted model.

**Table 1.** Comparison of deceased Canadian Longitudinal Study on Aging participants with complete and incomplete decedent questionnaires, 2012-2022s.

| Category | Characteristic | Completed Decedent Interview | Did Not Complete the Decedent Interview |
|---|---|---|---|
| | **Total Deceased** | **n = 1,287** | **n = 2,385** |
| **Sex** | Female | 489 (38.0%) | 974 (59.2%) |
| | Male | 798 (62.0%) | 1411 (40.8%) |
| **Age** | 45–64 | 115 (8.9%) | 562 (23.6%) |
| | 65–74 | 313 (24.3%) | 579 (24.3%) |
| | 75+ | 858 (66.7%) | 1,244 (52.1%) |
| **Ethnicity** | Non-White | 24 (1.8%) | 72 (3.0%) |
| | White | 1,263 (98.2%) | 2,313 (97.0%) |
| **Religion** | Not Religious | 227 (17.6%) | 720 (30.2%) |
| | Religious | 1,060 (82.4%) | 1,665 (69.8%) |
| **Education** | Less than High School | 170 (13.2%) | 357 (15.0%) |
| | High School | 158 (12.3%) | 355 (14.9%) |
| | Other post-secondary education | 507 (39.4%) | 1,008 (42.3%) |
| | University degree or above | 452 (35.1%) | 665 (27.8%) |
| **Marital Status** | Single, Divorced or Separated | 219 (17.0%) | 535 (22.4%) |
| | Married | 807 (62.7%) | 1,275 (53.5%) |
| | Widowed | 261 (20.3%) | 575 (24.1%) |

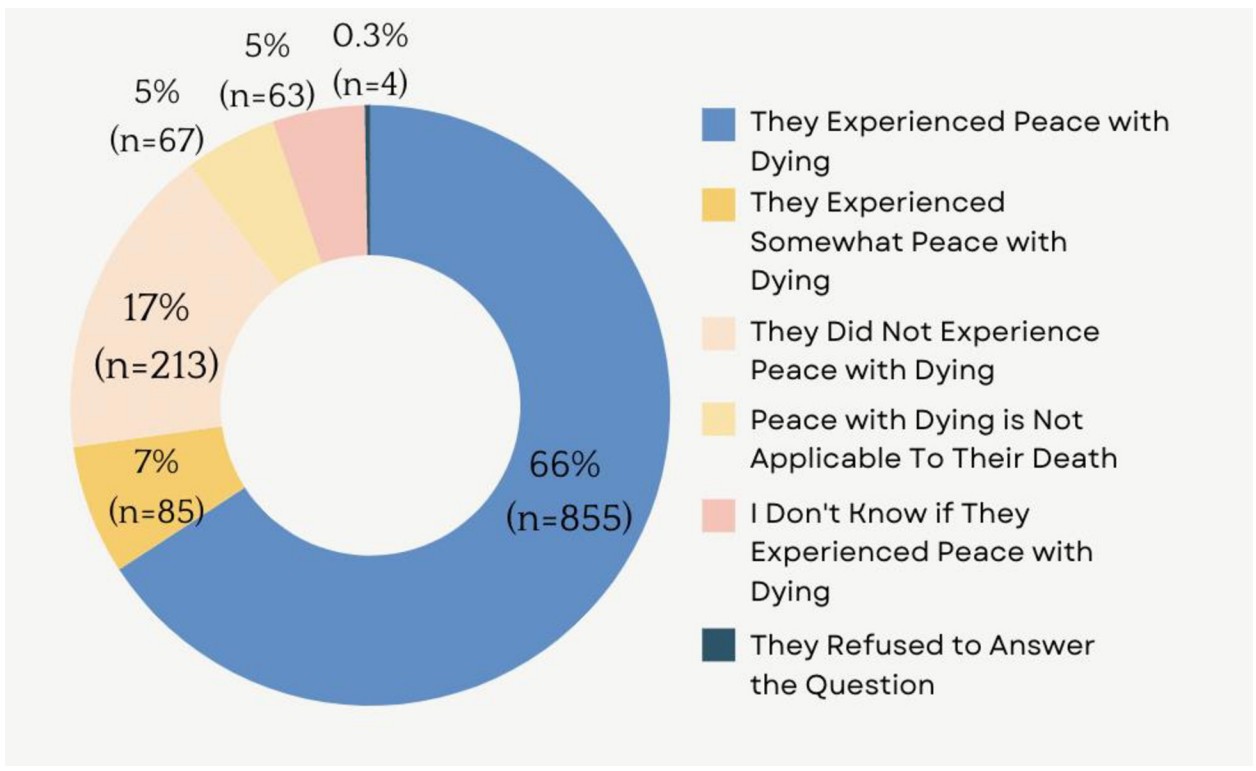

**Fig 1. Peace with dying among participants, the Canadian Longitudinal Study on Aging, 2012–2021.**

## Subgroup & sensitivity analysis

Stratified analyses showed similar end-of-life characteristics for females and males (**S4 Table**). Stratified models for cancer and non-cancerous cause of death found that persons dying of cancer had greater odds of peace with dying with an healthcare and end-of-life SDM, or when they pass away at home. Conversely, persons not dying of cancer had greater odds of peace with dying when they had physical impairment and the presence of an appointed end-of-life SDM (**S5 Table**). Sensitivity analysis comparing the unadjusted odds between interviews completed within one year of passing compared to between one to 6.3 years after passing showed similar results (**S1 Fig** and **S6 Table**). However, being male increased the odds of peace with dying when the interview was conducted within the first year after death and there was a lower likelihood of peace with dying after one year. Another sensitivity analysis examined the relationship of the respondent to the deceased participants (spouse, child or other) and its influence on the results and found no significant differences between the models (**S7 Table**).

## Discussion

In this Canadian study of older adults, we found that although most Canadians may experience peace with dying, close to 1 in 5 may not. This finding highlights the need for greater public attention and healthcare focus on improving the experience of dying. We documented that proxy reports of peace with dying are associated with a combination of personal characteristics, aspects of social connection, predictability of the illness trajectory, and end-of-life planning. Adjusted associations show that individuals who are widowed, diagnosed with cancer, and have an end-of-life SDM, were reported to have a higher likelihood of experiencing peace with dying. The interplay between cancer diagnosis, widowhood, and the presence of a decision maker

**Table 2. Unadjusted and adjusted odds of experiencing peace with dying, Canadian Longitudinal Study on Aging decedents, 2017–2022.**

| | Variable Category | n(%) | Experienced Peace with Dying (n = 855) | | | |
|---|---|---|---|---|---|---|
| | | | Unadjusted OR | 95% CI | Adjusted OR | 95% CI |
| **Sex** | Female | 798 (62.0%) | - | | - | |
| | Male | 489 (38.0%) | 0.93 | 0.73–1.18 | 1.06 | 0.80–1.39 |
| **Age** | 45–64 | 116 (9.0%) | - | | - | |
| | 65–74 | 313 (24.3%) | 1.27 | 0.82–1.96 | 1.15 | 0.80–1.66 |
| | 75+ | 858 (66.7%) | 1.55 | 1.04–2.30 | 1.30 | 0.91–1.87 |
| **Ethnicity** | Non-White | 24 (1.8%) | - | | - | |
| | White | 1,263 (98.2%) | 1.42 | 0.61–3.21 | 1.28 | 0.53–3.00 |
| **Religion** | Not Religious | 227 (17.6%) | - | | - | |
| | Religious | 1060 (82.4%) | 1.29 | 0.95–1.73 | 1.23 | 0.90–1.69 |
| **Education** | Less than High School | 170 (13.2%) | - | | - | |
| | High School | 158 (12.3%) | 0.98 | 0.61–1.57 | 1.02 | 0.63–1.66 |
| | Other post-secondary education | 507 (39.3%) | 0.86 | 0.59–1.24 | 0.85 | 0.57–1.26 |
| | University degree or above | 452 (35.1%) | 0.81 | 0.55–1.18 | 0.82 | 0.55–1.23 |
| **Marital Status** | Married | 807 (62.7%) | - | | - | |
| | Single / Divorced / Separated | 219 (17.0%) | 0.73 | 0.54–1.00 | 0.79 | 0.52–1.20 |
| | Widow | 261 (20.3%) | 1.53 | 1.12–2.10 | 1.51 | 1.01–2.29 |
| **ADL & IADL\*** | No/Mild Impairment | 464 (36.5%) | - | | - | |
| | Moderate impairment | 167 (13.0%) | 1.66 | 1.13–2.48 | 1.40 | 0.93–2.13 |
| | Severe/Total Impairment | 527 (40.9%) | 1.32 | 1.01–1.71 | 1.06 | 0.79–1.43 |
| **Caregiver** | Child | 362 (28.1%) | - | | - | |
| | Other | 336 (26.1%) | 1.13 | 0.77–1.67 | 1.18 | 0.83–1.68 |
| | Spouse | 589 (45.8%) | 1.32 | 1.01–1.86 | 1.02 | 0.70–1.49 |
| **Health Decision-Making SDM\*\*** | Absent | 238 (18.5%) | - | | - | |
| | Present | 1049 (81.5%) | 1.63 | 1.22–2.18 | 1.01 | 0.76–1.56 |
| **End-of-Life Decision-Making SDM\*\*** | Absent | 320 (24.9%) | - | | - | |
| | Present | 967 (75.1%) | 1.80 | 1.39–2.33 | 1.58 | 1.14–2.17 |
| **Closeness** | Not Close to Deceased | 109 (8.5%) | - | | - | |
| | Close to Deceased | 1178 (91.5%) | 1.38 | 0.92–2.05 | 1.22 | 0.79–1.87 |
| **Last physician visit** | Did Not See Doctor Before Passing | 715 (55.6%) | - | | - | |
| | 1–2 weeks | 176 (13.7%) | 1.08 | 0.76–1.55 | 1.07 | 0.73–1.57 |
| | 3–6 Weeks | 136 (10.6%) | 0.85 | 0.58–1.26 | 0.82 | 0.54–1.25 |
| | 7–51 Weeks | 133 (10.3%) | 0.64 | 0.44–0.94 | 0.64 | 0.42–0.96 |
| | 52+ Weeks | 127 (9.9%) | 0.77 | 0.52–1.14 | 0.83 | 0.54–1.29 |
| **Cause of death** | Heart Disease | 322 (25.0%) | - | | - | |
| | Cancer | 511 (39.7%) | 1.71 | 1.27–2.30 | 1.67 | 1.19–2.23 |
| | Other | 305 (23.7%) | 1.16 | 0.84–1.61 | 1.19 | 0.85–1.70 |
| | RIDK\*\* | 149 (11.6%) | 0.83 | 0.56–1.24 | 0.75 | 0.49–1.16 |
| **Location of Death** | Hospital | 631 (49.0%) | - | | - | |
| | Home | 292 (22.7%) | 1.16 | 0.87–1.56 | 0.79 | 0.57–1.09 |
| | Hospice/Palliative Care | 222 (17.2%) | 1.67 | 1.19–2.37 | 1.04 | 0.67–1.61 |
| | Senior Home/LTC[1]/Other | 142 (11.0%) | 1.17 | 0.80–1.73 | 1.07 | 0.67–1.72 |

*ADL/IADL = Activities of Daily Living/ Instrumental Activities of Daily Living

**SDM = Substitute Decision Maker

***RIDK = R = Respiratory diseases including emphysema, obstructive lung disease, asthma, chronic obstructive pulmonary disease; I = Influenza or pneumonia;

D = Dementia; K = Kidney Diseases such as nephritis, nephrotic syndrome, or nephrosis

[1]LTC = Long-term Care

highlights the complex interplay of personal circumstances, and preparedness or comfort with death, that can enhance the likelihood of experiencing peace with dying [9, 22].

Previous studies have found that achieving peace with dying inevitably results in a "good death" [23]. Our findings are consistent with previous shows a majority of older residents with experience peace with dying, however, quality of care and understanding of terminal diagnoses play a crucial role in peaceful end-of-life experiences [10, 15]. Older adults with known terminal diagnoses, such as cancer, are more likely to die in palliative care and have their wishes met before dying [24]. Individuals in this study diagnosed with cancer were more likely to experience peace with dying compared to those with other causes of death. A previous qualitative study found that individuals experience peace with dying when they have sufficient information about their disease and potential end-of-life care choices [9]. Many persons diagnosed with cancer have established health and end-of-life care plans and better access to healthcare attention than others, which may contribute to their sense of peace with dying.

The matter of dying plays an important role in quality of death and dying, as those who experience death closely may be more likely to experience peace with dying [25]. Widowed individuals compared to those who are married, experience death very closely, may have already navigated the complexities of grief and loss while reflecting on their experiences, making them more likely to have accepted death [26, 27]. Given the loss of a prominent social relationship and not being distressed about leaving their partners behind may allow them the ability to confront their own death more peacefully [28]. Although being surrounded by loved ones can alleviate feelings of loneliness and isolation, leading to a more comforting and peaceful end-of-life experience [2], the dual role of being both a patient and a widow may foster a unique understanding arising from their grief or loss of a loved one that may result in increasing their likelihood of experiencing peace with dying. Widows often reflect on their lives and relationships in their final days, similar to cancer survivors who see their suffering as a catalyst for personal growth and transformation, which brings a sense of peace [29].

Previous studies have reported fewer than 50% of participants having an appointed SDM [30]. We found over 75% of deceased CLSA participants with a decedent interview had a health and end-of-life SDM. We found that individuals with advance care directives or advance care planning documentations were more likely to experiencing peace with dying [24]. This suggests that establishing end-of-life plans may help alleviate emotional burden and fosters a sense of peace in individuals, leading to a more peaceful and improved quality of death [31, 32]. These findings support end-of-life planning initiatives such as, fulfilling individual's 'wishes' prior to dying, which brings forth a sense of peace with dying, inherently dignifying the dying process [33]. Our findings support the general presumption of advanced planning to support the quality of death [8, 34].

## Limitations

Though the CLSA decedent interviews were comprehensive, certain factors could not be considered, such as details of family support in the final months, quality of end-of-life care received, whether and how psychosocial-spiritual needs were fulfilled, and the level of comfort with the medical team [13, 35]. We did not provide respondents with an explicit definition for a peaceful death and each response was subject to the respondents' recall of the death.

The majority of participants from the sample are white and only represent a third of the entire decedent population. The CLSA is comparable to the 2011 Canadian census: as such decedents in this study were relatively young and white hence our sample may not represent the dying experience of the 'oldest old' or of other ethnicities. Although ethnicity-based census data in 2011 is not available, in a study examining end-of-life care between 2004–2012 in

Ontario, only 3.2% of total deaths were attributed to Chinese and South Asian ethnicities, but these groups had different end-of-life experiences [36]. Although our data could not capture these differences, this is the best available data we have on end-of-life experiences for a majority of older adults dying in Canada who have similar ethnic and religious backgrounds to our population. Therefore, as the largest study available to examine factors associated with peace with dying at national level, these findings are applicable to most end-of-life experiences in Canada.

Previous studies have identified that there is moderate agreement between family members on their assessment of peace with dying [37]. Therefore, this study is limited to one perspective and interviewing more than one respondent would have allowed for more perspectives of the dying experience. Similarly, caregivers may overestimate the degree of pain, and other symptoms, compared to the individual experiencing the symptoms [38, 39]. Receiving patient perspectives of the quality of death and dying may have provided a more accurate representation of the overall dying experience, which we did not have access to in this study. Finally, we did not have demographic or care provider description on the decedent respondent, however, this is the best available data to understand CLSA participants end-of-life experience based on various respondent perspectives and varying degrees of closeness to the deceased (Table 2 and S7 Table).

Although limited, our study is susceptible to potential biases associated with the time elapsed after an individual's passing and the family members' reporting and or the completion of the decedent interview. It is conceivable that over time, the recollection of events may be influenced by various factors, potentially leading to a bias favoring positive end-of-life experiences.

## Conclusions and implications

Though most older Canadians may experience peace with dying, many may not. Experiencing peace with dying is multifaceted, influenced by a combination of personal characteristics, end-of-life planning, access to end-of-life care, and predictability of the illness trajectory. Awareness of factors that are associated with peaceful dying experiences may allow tailored interventions to better meet individual's needs, facilitating dignified, comfortable end-of-life experiences for older adults.

## Supporting information

**S1 Fig. Histogram of time to interview after participant death, Canadian Longitudinal Study on Aging, 2012–2022.**
(PDF)

**S1 Table. Characteristics of deceased Canadian Longitudinal Study on Aging participants with a completed decedent interview (2012–2022) compared to Canadian decedent population (2012–2022).**
(PDF)

**S2 Table. Correlation matrix for all participant characteristics and end-of-life characteristics, Canadian Longitudinal Study on Aging, 2012–2021.**
(PDF)

**S3 Table. Variance inflation factor for all participant characteristics and end-of-life characteristics, Canadian Longitudinal Study on Aging, 2012–2021.**
(PDF)

**S4 Table. End-of-life characteristics for males and females with completed decedent interviews, Canadian Longitudinal Study on Aging, 2012–2022.**
(PDF)

**S5 Table. Subgroup analysis of Canadian Longitudinal Study on Aging deceased participants with a completed decedent interview based on cause of death, 2012–2022.**
(PDF)

**S6 Table. Sensitivity analysis of peace with dying for participants with completed decedent interview within one year of death compared to more than one year after death, Canadian Longitudinal Study on Aging, 2012–2022.**
(PDF)

**S7 Table. Sensitivity analysis of peace with dying for participants with different decedent respondents, Canadian Longitudinal Study on Aging, 2012–2022.**
(PDF)

## Acknowledgments

Declarations

The opinions expressed in this manuscript are the author's own and do not reflect the views of the Canadian Longitudinal Study on Aging or the Government of Canada.

## Author Contributions

**Conceptualization:** Komal Aryal, Susan Kirkland, Andrew P. Costa.

**Data curation:** Komal Aryal, Andrew P. Costa.

**Formal analysis:** Komal Aryal, Aaron Jones, Peter Tanuseputro, Lauren E Griffith, Andrew P. Costa.

**Funding acquisition:** Andrew P. Costa.

**Investigation:** Komal Aryal, Paul C. Hebert, Deborah J. Cook.

**Methodology:** Komal Aryal.

**Project administration:** Komal Aryal.

**Resources:** Komal Aryal, Deborah J. Cook.

**Software:** Komal Aryal, Andrew P. Costa.

**Supervision:** Andrew P. Costa.

**Validation:** Komal Aryal, Peter Tanuseputro, Susan Kirkland.

**Visualization:** Komal Aryal, Aaron Jones, Paul C. Hebert, Susan Kirkland.

**Writing – original draft:** Komal Aryal.

**Writing – review & editing:** Komal Aryal, Aaron Jones, Peter Tanuseputro, Lauren E Griffith, Paul C. Hebert, Susan Kirkland, Deborah J. Cook, Andrew P. Costa.

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
