## [Decision Letter · Decision Letter 0]

26 Nov 2024

PONE-D-24-48128Peaceful Dying Among Canada's Elderly: An Analysis of the Canadian Longitudinal Study on AgingPLOS ONE

Dear Dr. Aryal,

Thank you for submitting your manuscript to PLOS ONE. After careful consideration, we feel that it has merit but does not fully meet PLOS ONE’s publication criteria as it currently stands. Therefore, we invite you to submit a revised version of the manuscript that addresses the points raised during the review process.

We look forward to receiving your revised manuscript.

Kind regards,

Mario Ulises Pérez-Zepeda, M.D., Ph.D.

Academic Editor

PLOS ONE

**Journal Requirements:**

Funding for this study was obtained from the Canada Research Chair in Integrated Care for Seniors.

Reviewers' comments:

Reviewer's Responses to Questions

**Comments to the Author**

1. Is the manuscript technically sound, and do the data support the conclusions?

Reviewer #1: Yes

Reviewer #2: Partly

2. Has the statistical analysis been performed appropriately and rigorously? 

Reviewer #1: Yes

Reviewer #2: Yes

3. Have the authors made all data underlying the findings in their manuscript fully available?

Reviewer #1: Yes

Reviewer #2: No

4. Is the manuscript presented in an intelligible fashion and written in standard English?

Reviewer #1: Yes

Reviewer #2: Yes

5. Review Comments to the Author

**Reviewer #1: **I was a pleasure to review this paper, I found it very innovative and relevant. Though, I do have some comments meant only to improve your work.

In the methods section it would be helpful to clarify the adjustment variables, and why are they relevant for this study. In the results section, it would be helpful to also have a table for the adjusted model. Finally in the discussion section, it would be helpful to have some information (if it exists) on peaceful deaths in other countries or age groups, just to help the reader understand whether 20% is a high number. Particularly, since Canada has universal healthcare with access to palliative care (overall), hospice and even MAID.

Thank you

**Reviewer #2: **Thank you for the opportunity to review this fine and important paper. While much is written about the challenges implicit in approaching death, less is said about normal dying and degree to which people can approach end of life with a sense of calm or peace and the absence of much feared suffering.

The data is drawn from a very impressive longitudinal study, and hence provides an opportunity for a large population-based examination of a topic that is usually broached with either case reports, case series, and smaller cohorts than the one CSLA provides.

A few more substantive comments, followed by a series of minor comments regarding primarily issues of clarification.

1. I’m assuming that you have no additional data on the circumstances surrounding the patient’s death (quality and nature of care they received; any neasure(s) of their suffering/symptom distress/peace, etc.); this would, of course, strengthened the reported findings, given the current paper is based on proxy reports by surviving family contacts.

2. The primary finding is that the majoring of patients died peacefully. Very little is said about this in the discussion, other than that 1/5 don’t. While the latter point is well taken - and we need to understand why that is the case and how that might inform our ability to provide better palliative care - some discussion about the prominence of peaceful dying is important. So many people are afraid of death and anticipate overwhelming suffering/distress. A methodologically sound study, painting a more complete picture, is important to expound on and how this insight provides the ability to counsel patients and families encountering end-of-life circumstances.

3. We are provided almost no information about the participants who provided data regarding the deceased patient. Some description, and perhaps even a table, including basic demographics (gender/age/relationship with deceased/proximity to role of care provider/any other basic information that might have been collected on these participants) would be helpful.

Here are some minor issues/observations in the order they appear in the paper:

Line 146: “has been linked not experiencing peace with dying” should read “has been linked to not experiencing peace with dying

Line 215: a comment … it is perhaps no surprise that decedents were more likely to be male and married, given their female partners live longer; and having been married means they are more likely to have someone survive them willing to take part in the decent interview.

Line 229: I believe this is meant to read, “Almost two-thirds (62.0%, n=798) of deceased participants were male”

Line 233: Readers may not understand the distinction between end-of-life substitute decision makers, and health care substitute decision makers.

Line 253: There is something wrong with this sentence: “similar for experiencing peace with dying model was (AUC=0.65).

Line 286: “potential end-of-life care pathways” may be a bit obtuse for some readers; I think you are wanting to say that patients do better when they understand and are given sufficient information about their likely disease course

Line 290: This suggests that “people who experience death more closely” are more at peace with dying. I’m not certain this generalization is accurate, as it often depends on the circumstances/quality/nature of the death they have observed. Witnessing a traumatic course of dying can taint and profoundly distort how people anticipate their own death. Reference 24 that is provided doesn’t mention the topic of peaceful death or observations of death more closely.

Line 291-299: This deals with the issue of why widows seem to be ‘more at peace’ with death. The arguments are speculative and should be framed as such. For instance, although widows may experience a death up close, as previously stated, that does not result in equanimity in approaching their own death (again, much may depend on what they observed and whether it provides a reassuring or a terrifying templated – or something in between – for what their death might look like). It would be helpful if you could embed your speculation in some references that deal more specifically with this issue.

It is later stated that widows reflect on their lives in their final days, and that like patients with cancer, this promotes personal growth and transformation. Reference 25 however does not mention widows, making the argument seem less convincing.

Another possibility worth considering is the matter of dying in the absence of a spouse/intimate partner to bear witness. There is much to suggest that feeling a burden to others is a powerful dynamic in shaping end-of-life experience. This paper (Chochinov HM et al. The landscape of distress in the terminally ill. J Pain Symptom Manage. 2009;38:641-9) also intriguingly reports that not having a partner/living alone seems to mitigate dignity related distress and provides speculation on why that might be the case. This may be of interest as you revisit how to help readers grapple with understanding the ‘widows are more at peace’ data.

In response to the questions the journal asks reviewers to address, my response to all of them is a definitive yes.

1. The study presents the results of original research.

2. Results reported have not been published elsewhere.

3. Experiments, statistics, and other analyses are performed to a high technical standard and are described in sufficient detail.

4. Conclusions are presented in an appropriate fashion and are supported by the data.

5. The article is presented in an intelligible fashion and is written in standard English.

6. The research meets all applicable standards for the ethics of experimentation and research integrity.

7. The article adheres to appropriate reporting guidelines and community standards for data availability.

Once the revisions are attended to, I look forward to seeing this paper published. It is an important contribution to our understanding of how older patients die.

6. PLOS authors have the option to publish the peer review history of their article (what does this mean?). If published, this will include your full peer review and any attached files.

Reviewer #1: No

Reviewer #2: **Yes: **Dr Harvey Max Chochinov

---

## [Author Response · Author response to Decision Letter 0]

5 Dec 2024

Reviewer #1: 

I was a pleasure to review this paper, I found it very innovative and relevant. Though, I do have some comments meant only to improve your work.

Response: Thank you for your kind comment. I will address your other comments below.

In the methods section it would be helpful to clarify the adjustment variables, and why are they relevant for this study. In the results section, it would be helpful to also have a table for the adjusted model. 

Response: Thank you for this suggestion. In line 164 we have reported all the variable categories we have included in both the adjusted and unadjusted models: “We examined participant and end-of-life characteristics from the decedent questionnaire, including the location of death, cause of death, arrangements for health care decision-making, and arrangements for end-of-life care decision-making.” 

To line 174, we clarified why we included the adjusted model: “We performed unadjusted logistic regression to estimate associations with peace with dying and multivariable logistic regression to determine adjusted associations and adjust for any potential confounders.” 

The unadjusted and adjusted model results are presented in Table 2 in the results section of the manuscript. 

Finally in the discussion section, it would be helpful to have some information (if it exists) on peaceful deaths in other countries or age groups, just to help the reader understand whether 20% is a high number. Particularly, since Canada has universal healthcare with access to palliative care (overall), hospice and even MAID.

Response: We have cited two studies, one from the Netherlands and one from the US, on line 267 that demonstrate that peace with dying amongst older adults is similar to our findings. We have also explained these findings in lines 265. 

Reviewer #2: 

Thank you for the opportunity to review this fine and important paper. While much is written about the challenges implicit in approaching death, less is said about normal dying and degree to which people can approach end of life with a sense of calm or peace and the absence of much feared suffering. 

Response: Thank you for your comment. The objective of this paper was to understand factors associated with peace with dying. We wanted to understand the individuals characteristics that allow some older adults to experience peace with dying. We did not have data available on approaches that individuals may have taken to experience a peaceful death, which we have addressed in the limitations. We have commented on the findings available in the data in order to meet our objectives. 

The data is drawn from a very impressive longitudinal study, and hence provides an opportunity for a large population-based examination of a topic that is usually broached with either case reports, case series, and smaller cohorts than the one CSLA provides.

A few more substantive comments, followed by a series of minor comments regarding primarily issues of clarification.

Response: Thank you for your kind comment about the data. 

1. I’m assuming that you have no additional data on the circumstances surrounding the patient’s death (quality and nature of care they received; any measure(s) of their suffering/symptom distress/peace, etc.); this would, of course, strengthened the reported findings, given the current paper is based on proxy reports by surviving family contacts.

Response: Unfortunately, we do not have additional data on the circumstances surrounding participants death beyond what is presented in the study. We have stated and clarified this in the limitations on line 309 “Though the CLSA decedent interviews were comprehensive, certain factors could not be considered, such as details of family support in the final months, quality of end-of-life care received, whether and how psychosocial-spiritual needs were fulfilled, and the level of comfort with the medical team.” 

2. The primary finding is that the majoring of patients died peacefully. Very little is said about this in the discussion, other than that 1/5 don’t. While the latter point is well taken - and we need to understand why that is the case and how that might inform our ability to provide better palliative care - some discussion about the prominence of peaceful dying is important. So many people are afraid of death and anticipate overwhelming suffering/distress. A methodologically sound study, painting a more complete picture, is important to expound on and how this insight provides the ability to counsel patients and families encountering end-of-life circumstances.

Response: Thank you for this comment. We agree that addressing those who experience peace with dying is important. To address this we have added the following statements to lines 265: “Our findings are consistent with previous shows a majority of older residents with experience peace with dying, however, quality of care and understanding of terminal diagnoses play a crucial role in peaceful end-of-life experiences. Older adults with known terminal diagnoses, such as cancer, are more likely to die in palliative care and have their wishes met before dying.” 

3. We are provided almost no information about the participants who provided data regarding the deceased patient. Some description, and perhaps even a table, including basic demographics (gender/age/relationship with deceased/proximity to role of care provider/any other basic information that might have been collected on these participants) would be helpful.

Response: We have no additional information on the decedent respondent besides how close they were to the deceased participant (included in the unadjusted and adjusted models) and the relationship of the respondent to the deceased participant. We have clarified this in our limitations section line 333-336: “Finally, we did not have demographic or care provider description on the decedent respondent, however, this is the best available data to understand CLSA participants end-of-life experience based on various respondent perspectives and varying degrees of closeness to the deceased (Table 2 and Appendix H).” 

We found no significant differences between the models in our sensitivity analysis on the relationship of the respondent to the deceased participants (spouse, child or other) and its influence on the results. 

Here are some minor issues/observations in the order they appear in the paper:

Line 146: “has been linked not experiencing peace with dying” should read “has been linked to not experiencing peace with dying

Response: Thank you for your comment, we have made this fix and the statement reads “has been linked to not experiencing peace with dying.”

Line 215: a comment … it is perhaps no surprise that decedents were more likely to be male and married, given their female partners live longer; and having been married means they are more likely to have someone survive them willing to take part in the decent interview.

Response: Thank you for your comment. We reported these findings to show the general demographics of the deceased population. 

Line 229: I believe this is meant to read, “Almost two-thirds (62.0%, n=798) of deceased participants were male”

Response: Thank you for reviewing this. I have corrected this to “Almost two-thirds (62.0%, n=798) of deceased participants were male.”

Line 233: Readers may not understand the distinction between end-of-life substitute decision makers, and health care substitute decision makers.

Response: Thank you for your comment. We have clarified the roles of each substitute decision maker alongside the results: “Regarding end-of-life characteristics, 39.7% (n=511) died of cancer, 49% (n=631) died in hospital, and 75.1% (n=967) and 81.5% (n=1,049) of participants had appointed proxies responsible for deciding any end-of-life choices known as end-of-life substitute decision makers (SDM) and health care SDMs, proxies responsible for broader healthcare decisions, respectively.” 

Line 253: There is something wrong with this sentence: “similar for experiencing peace with dying model was (AUC=0.65).

Response: Thank you for reviewing this. I have revised the sentence so it reads as follows “Model discrimination for the adjusted model was fair (AUC=0.65) and similar to the unadjusted model.”

Line 286: “potential end-of-life care pathways” may be a bit obtuse for some readers; I think you are wanting to say that patients do better when they understand and are given sufficient information about their likely disease course

Response: Thank you for your comment and we understand the confusion. We have reworded the sentence as such: “A previous qualitative study found that individuals experience peace with dying when they have sufficient information about their disease and potential end-of-life care choices.”

Line 290: This suggests that “people who experience death more closely” are more at peace with dying. I’m not certain this generalization is accurate, as it often depends on the circumstances/quality/nature of the death they have observed. Witnessing a traumatic course of dying can taint and profoundly distort how people anticipate their own death. Reference 24 that is provided doesn’t mention the topic of peaceful death or observations of death more closely.

Response: The paper cited indicates that surviving caregivers have better bereavement outcomes, when their loved ones have better quality of death outcomes, such as peace with dying. We have rephrased the sentence so it suggests less of a direct relationship “The matter of dying plays an important role in quality of death and dying, as those who experience death closely may be more likely to experience peace with dying.”

Line 291-299: This deals with the issue of why widows seem to be ‘more at peace’ with death. The arguments are speculative and should be framed as such. For instance, although widows may experience a death up close, as previously stated, that does not result in equanimity in approaching their own death (again, much may depend on what they observed and whether it provides a reassuring or a terrifying templated – or something in between – for what their death might look like). It would be helpful if you could embed your speculation in some references that deal more specifically with this issue. It is later stated that widows reflect on their lives in their final days, and that like patients with cancer, this promotes personal growth and transformation. Reference 25 however does not mention widows, making the argument seem less convincing.

Another possibility worth considering is the matter of dying in the absence of a spouse/intimate partner to bear witness. There is much to suggest that feeling a burden to others is a powerful dynamic in shaping end-of-life experience. This paper (Chochinov HM et al. The landscape of distress in the terminally ill. J Pain Symptom Manage. 2009;38:641-9) also intriguingly reports that not having a partner/living alone seems to mitigate dignity related distress and provides speculation on why that might be the case. This may be of interest as you revisit how to help readers grapple with understanding the ‘widows are more at peace’ data.

Response: Thank you for your suggestion. We agree with the reviewer that “although widows may experience a death up close, as previously stated, that does not result in equanimity in approaching their own death,” and have expanded on this argument. We have added literature by Park and Costa which describes how widowed spouses may have already navigated the complexities of grief, loss, and reflection allowing them to be more comfortable with death on lines 277-280. This better explains how widows would be more at peace with dying compared to those who are married. 

We have also expanded our argument using the paper cited by the reviewer to expand on widowhood increasing the likelihood of experiencing peace with dying on line 280-282: “Given the loss of a prominent social relationship and not being distressed about leaving their partners behind may allow them the ability to confront their own death more peacefully.” We believe this added literature explains why widowed individuals are more likely to experience peace with dying. 

In response to the questions the journal asks reviewers to address, my response to all of them is a definitive yes.

1. The study presents the results of original research.

2. Results reported have not been published elsewhere.

3. Experiments, statistics, and other analyses are performed to a high technical standard and are described in sufficient detail.

4. Conclusions are presented in an appropriate fashion and are supported by the data.

5. The article is presented in an intelligible fashion and is written in standard English.

6. The research meets all applicable standards for the ethics of experimentation and research integrity.

7. The article adheres to appropriate reporting guidelines and community standards for data availability.

Once the revisions are attended to, I look forward to seeing this paper published. It is an important contribution to our understanding of how older patients die.

Response: Thank you for supporting this work and providing all your comments.

---

## [Editor Report · Decision Letter 1]

20 Dec 2024

Peaceful Dying Among Canada's Elderly: An Analysis of the Canadian Longitudinal Study on Aging

PONE-D-24-48128R1

Dear Dr. Aryal,

We’re pleased to inform you that your manuscript has been judged scientifically suitable for publication and will be formally accepted for publication once it meets all outstanding technical requirements.

Kind regards,

Mario Ulises Pérez-Zepeda, M.D., Ph.D.

Academic Editor

PLOS ONE
---

## [Editor Report · Acceptance letter]

9 Jan 2025

PONE-D-24-48128R1 

PLOS ONE

Dear Dr. Aryal, 

I'm pleased to inform you that your manuscript has been deemed suitable for publication in PLOS ONE. Congratulations! Your manuscript is now being handed over to our production team.

Kind regards, 

on behalf of

Dr. Mario Ulises Pérez-Zepeda 

Academic Editor

PLOS ONE